

# Prediction Error Growth in a more Realistic Atmospheric Toy Model with Three Spatiotemporal Scales

Hynek Bednář[1,2] and Holger Kantz[1]

[1]Max Planck Institute for the Physics of Complex Systems (MPIPKS), D-01187, Dresden, Germany
[2]Department of Atmospheric Physics, Faculty of Mathematics and Physics, Charles University, 18000, Prague, Czech Republic

*Correspondence to*: Hynek Bednář (hynek.bednar@mff.cuni.cz)

**Abstract.** This article studies the growth of the prediction error over lead time in a schematic model of atmospheric transport. Inspired by the Lorenz (2005) system, we mimic an atmospheric variable in 1 dimension, which can be decomposed into three spatiotemporal scales. We identify parameter values that provide spatiotemporal scaling and chaotic behavior. Instead of exponential growth of the forecast error over time, we observe a more complex behavior. We test a power law and the quadratic hypothesis for the scale dependent error growth. The power law is valid for the first days of the growth, and with an included saturation effect, we extend its validity to the entire period of growth. The theory explaining the parameters of the power law is confirmed. Although the quadratic hypothesis cannot be completely rejected and could serve as a first guess, the hypothesis's parameters are not theoretically justifiable. In addition, we study the initial error growth for the ECMWF forecast system (500 hPa geopotential height) over the 1986 to 2011 period. For these data, it is impossible to assess which of the error growth descriptions is more appropriate, but the extended power law, which is theoretically substantiated and valid for the Lorenz system, provides an excellent fit to the average initial error growth of the ECMWF forecast system. Fitting the parameters, we conclude that there is an intrinsic limit of predictability after 22 days.

## 1. Introduction

The improvement of the numerical weather prediction systems raised the question of the intrinsic atmospheric prediction limit, i.e., for the maximal lead time into the future, after which every forecast will be useless. While the notion of seamless prediction (Shukla, 2009) and of seasonal prediction implies that it will be only a matter of technology to make forecasts far into the future, in recent years, there have been several publications whose authors assume a strict upper bound in time for making useful predictions (Brisch and Kantz, 2019; Zhang et al.,2019, Palmer et al., 2014). Even if the numerical model were perfect, the uncertainty of the initial condition would give rise to prediction errors which grow over time. In the setting of classical low dimensional chaos, one would observe an exponential error growth whose exponent is given by the largest Lyapunov exponent of the system, with some saturation when the error reaches the magnitude of the standard deviation of the quantity to be predicted. Although exponential error growth has been associated with the fact that a detailed forecast is meaningful only up





to lead times of a few multiples of the Lyapunov time, which is the inverse of the Lyapunov exponent, in principle, with absolutely perfect knowledge of the initial condition, one could compute meaningful predictions up to arbitrary times.

In contrast to this, Brisch and Kantz (2019) and Zhang et al. (2019) predicted a strictly finite prediction horizon, which is associated with a scale dependent error growth, where tiny errors grow much faster than larger errors. In essence, this would mean that the proper Lyapunov exponent of the system was infinite, but that finite size approximations of the Lyapunov

exponent (Cencini, 2013) were the smaller, the larger the scale of this finite size.

Palmer et al. (2014), referring to Lorenz (1969), call this growth the "real butterfly effect," and it is not an exponential error growth defined by the largest positive Lyapunov exponent, but growth where the exponent is replaced by a scale dependent quantity (scale dependent error growth rate).

The atmosphere exhibits multi-scale dynamics both in space and in time, and observations show a close linkage between spatial

and temporal scales: the smaller some structure, the shorter its lifetime and the faster its time evolution. Planetary scale structures (e.g., semi-permanent pressure centers or the westerlies and trade winds) have sizes in the order of ten thousand km and live on time scales of weeks and longer. Synoptic scale structures (e.g., high and low pressure systems) have sizes of several thousands of km and live on time scales of several days. Mesoscale structures (e.g., thunderstorms and weather fronts) have sizes from a few kilometers to several hundred kilometers and live on a time scale of a day or less. Microscale structures

(e.g., turbulence) have sizes smaller than 1 km and live on a time scale of minutes. In addition to different spatiotemporal scales, structures also have different error growth rates and predictability. Error growth is faster, and predictability is less for smaller scale structures.

Lorenz (1996) gave a sketch of error growth in such a system: A typical quantity to be predicted is a superposition of the dynamics on different scales. After a fast growth of the small scale errors with saturation at these very same small scales, the

large scale errors continue to grow at a slower rate until even these saturate. Therefore, Lyapunov exponents of structures of various spatiotemporal scales are taken as the previously mentioned scale dependent quantity, and they determine the error growth on their respective scales.

Zhang et al. (2019) take a very different starting point and suggest the quadratic hypothesis to describe the scale dependent error growth. It was originally designed to describe initial and model error growth (Savijarvi, 1995; Dalcher and Kalney, 1987).

Zhang et al. (2019) newly used a parameter previously specifying the model error to describe upscale error propagation from small scale processes and showed the validity of this hypothesis on data of the numerical weather prediction systems of the European Centre for Medium Range Weather Forecasts (ECMWF) and the U.S. Next Generation Global Prediction System.

A scale dependent error growth in the spirit of Lorenz (1996) was described by Brisch and Kantz (2019) using a power law, which successfully approximated the data of the National Center for Environmental Protect Global Forecast System (Harlim

et al., 2005). Brisch and Kantz (2019) also introduced a theory connecting the power law exponent with the Lyapunov exponents and limit errors of the different scales, and its validity was demonstrated by a low-dimensional atmospheric system (Lorenz, 1996) extended to five spatiotemporal levels. However, the different scales in this model cannot be superimposed in





order to gain a general signal in the real space, but the different scales were living in different subspaces of phase space. This lack of realism may limit the general acceptance of this theory and stimulated the present study.

This article expands Lorenz's (2005) system from two spatiotemporal levels to three and discusses the setting of parameters and its advantage over other systems. In this system, the scale dependent growth of the initial error is calculated and is approximated by the power law and the quadratic hypothesis. The results are discussed, the power law is modified, and the theoretical justifications of the approximations' parameter values are sought and verified. The findings are then applied to the initial error growth of the European Centre for Medium-Range Weather Forecasts (ECMWF) numerical weather prediction

system over the 1986 to 2011 period (500 hPa geopotential height).

This article is divided into five sections. The second describes the system with three spatiotemporal scales based on Lorenz (2005). In Sec. 3, we present the numerical error growth behavior and fit it with previously suggested laws such as a power law growth and the so called quadratic law, where we also introduce extensions into the saturation regime at large errors. In Sec. 4, we trace back the empirically found parameters of the power law error growth to properties of the system and show

that we can explain these findings self-consistently with the different error growth rates at different scales. In the fifth section, we perform a similar analysis for data from the ECMWF forecast system. Conclusions and discussions are then presented in the final section.

## 2. Multi-hierarchical system L05-3

The designed system with three spatiotemporal levels (L05-3) is based on systems created by Lorenz (2005). The first and

simplest of this type is the low-dimensional atmospheric system (L96) presented by Lorenz (1996). It is a nonlinear model, with $N$ variables connected by governing equations

$$dX_n / dt = -X_{n-2}X_{n-1} + X_{n+1}X_{n-1} - X_n + F, \qquad (1)$$

$n = 1,\ldots,N$ . $X_{n-2}$ , $X_{n-1}, X_n$ , $X_{n+1}$ are unspecified (i.e., unrelated to actual physical variables) scalar meteorological quantities (units), $F$ is a constant representing external forcing, and $t$ is time. The index is cyclic so that $X_{n-N} = X_{n+N} = X_n$

and variables can be viewed as existing around a latitude circle. Nonlinear terms of Eq. (1) simulate advection. Linear terms represent mechanical and thermal dissipation. The model quantitatively, to a certain extent, describes weather systems, but, unlike the well-known Lorenz model of atmospheric convection (Lorenz, 1963), it cannot be derived from any atmospheric dynamic equations. The motivation was to formulate the simplest possible set of dissipative chaotically behaving differential equations that share some properties with the "real" atmosphere. One of the model's properties is to have 5 to 7 main highs

and lows that correspond to planetary waves (Rossby waves) and several smaller waves corresponding to synoptic-scale waves. For Eq. (1), this is only valid for $N = 30$ . Lorenz (2005), therefore, introduced spatial continuity modification (L05). Eq. (1) is then rewritten to the form:





$$\frac{dX_n}{dt} = [X,X]_{L,n} - X_n + F, \tag{2}$$

where


$$[X,X]_{L,n} = \sum_{j=-J}^{J}{}' \sum_{i=-J}^{J}{}' \left( -X_{n-2L-i}X_{n-L-j} + X_{n-L+j-i}X_{n+L+j} \right)/L^2$$

If $L$ is even, $\sum'$ denotes a modified summation, in which the first and last terms are to be divided by 2. If $L$ is odd, $\sum'$ denotes an ordinary summation. Generally, $L$ is much smaller than $N$ and $J = L/2$ if $K$ is even and $J = (L-1)/2$ if $L$ is odd. To keep a desirable number of main highs and lows, Lorenz (2005) suggested a ratio $N/L = 30$ and $F = 15$. The choice of parameters $F$, along with the setting of time unit = 5 days, is also made to obtain a similar value of the largest

Lyapunov exponent as the ECMWF forecasting system (Lorenz, 2005).

A two-level (scales) system (L96-2) was introduced by Lorenz (1996) by coupling two such systems, each of which, aside from the coupling, obeys a suitably scaled variant of Eq. (1). There are $N$ variables $X_n$ plus $JN$ variables $Y_{j,n}$ defined for $n = 1,\ldots,N$ and $j = 1,\ldots,J$. Governing equations are:

$$dX_n/dt = -X_{n-2}X_{n-1} + X_{n+1}X_{n-1} - X_n + F - (c/b)\sum_{j=1}^{J}Y_{j,n}, \tag{3}$$


$$dY_{j,n}/dt = -cbY_{j-2,n}Y_{j-1,n} + cbY_{j+1,n}Y_{j-1,n} - cY_{j,n} + (c/b)X_n, \tag{4}$$

where $c$ sets the rapidness of small scale compared to large scale, $b$ sets the small scale amplitude size compared to large scale. $Y_{j,n-N} = Y_{j,n+N} = Y_{j,n}$ while $Y_{j+J,n} = Y_{j,n+1}$ and $Y_{j-J,n} = Y_{j,n-1}$. $X_n$ represent the values of some quantity in $N$ sectors of latitude circle, while the variables $Y_{j,n}$ ( $Y_{1,1}, Y_{2,1},\ldots,Y_{J,1}, Y_{1,2}, Y_{2,2},\ldots,Y_{J,2}, Y_{3,1},\ldots$ ) can represent some other quantity in $JN$ sectors. In this system, one could construct a more general variable that is defined on all of the $JN$ sectors, namely

$Z_{j,k} = X_j + Y_{j,k}$.

Similarly to the L96-2 systems, Britch and Kantz (2019) created L levels systems (L96-H). Their model, however, lacks an essential property of atmospheric variables: The different levels of their hierarchy are different variables, occupying different subspaces of the phase space, as if it were different Fourier modes of some system, but were defined in real space. Therefore, we introduce here a model which is closer to reality.

We start from a system L05-2 like L96-2 (Lorenz, 2005):

$$dX_n/dt = [X,X]_{L,n} - X_n - cY_n + F, \tag{5}$$





$$dY_n / dt = b^2 [Y,Y]_{1,n} - bY_n + cX_n. \tag{6}$$

Eq. (6) is identical to Eq. (1), and Eq. (5) is identical to Eq. (2) (aside from the coupling where $c$ is the coupling coefficient, and that $Y_n$ fluctuates $b$ times as rapidly, and their amplitude is reduced by the factor $b$ ). L05-2 or L96-2 system, however, has an unrealistic property compared to the numerical weather prediction systems. The large-scale and small-scale features are represented by separate sets of variables $X$ and $Y$ instead of appearing as superimposed features of a single set $Z$ . Lorenz (2005) wanted to keep the system as simple as possible, so instead of, for example, Fourier analysis, a procedure for expressing variables $Z_n$ as sums of $X_n$ and $Y_n$ was introduced:

$$X_n = \sum_{i=-I}^{I}{}'(\alpha - \beta|i|)Z_{n+i}, \tag{7}$$

$$Y_n = Z_n - X_n. \tag{8}$$

Parameters $\alpha$ , $\beta$ , and $I$ are chosen so that $X$ is a low-pass filtered version of $Z$ , and $Y$ represents the difference between the full signal $Z$ and the filtered signal. By this procedure, $Y$ has a much smaller amplitude than $X$ , and also its time evolution should be faster since the temporal derivative is related to the spatial derivative via the difference $(X_{n+1} - X_{n-2})$ , which for the low pass filtered signal $X$ typically is smaller than for the signal $Y$ .

More precisely, Lorenz's (2005) idea is that the parameters $\alpha$ , $\beta$ are chosen so that $X$ equals $Z$ whenever $Z$ changes quadratically over the longitudes (variables) $n - I$ through $n + I$. It is when $\sum_{i=-I}^{I}{}'(\alpha - \beta|i|) = 1$ and $\sum_{i=-I}^{I}{}' i^2 (\alpha - \beta|i|) = 0$ . By solving these equations, we get:

$$\alpha = (3I^2 + 3)/(2I^3 + 4I) \tag{9}$$

$$\beta = (2I^2 + 1)/(I^4 + 2I^2). \tag{10}$$

The procedures (Eqs. (7) and (8)) are functions of the length of the interval $[-I, I]$. When creating a system $dZ / dt$ as the sum of $dX / dt$ and $dY / dt$ (sum of Eqs. (5) and (6)), the coupling term $cX_n$ in Eq. (6), which enables short waves to develop, is combined with the dissipation term $-X_n$ in Eq. (6). Therefore the coupling term can be completely canceled, or it can appear in $X$ rather than $Y$ when Z is analyzed, and there might be nothing to enable the short waves in $Y$ to grow. Lorenz (2005)





reformulated the coupling process by adding a small fraction of $X$ to $Y$ so small waves in $Y$ can amplify and proved that

this is done by replacing $b^2[Y,Y]_{1,n}+cX_n$ by $[Y,Y+c'X]_{1,n}$ in Eq. (6) and a new form of L05-2 system would be:

$$dZ_n / dt = [X,X]_{L,n} + b^2[Y,Y]_{1,n} + c[Y,X]_{1,n} - X_n - bY_n + F, \qquad (11)$$

where $c = c'b^2$.

Based on the L05-2 system (Eqs. (7) - (11)), We designed a three levels (scales) system (L05-3):

$$dX_{tot,n} / dt = [X_1,X_1]_{L,n} + b_1^2[X_2,X_2]_{1,n} + b_2^2[X_3,X_3]_{1,n} + c_1[X_2,X_1]_{1,n} + c_2[X_3,X_2]_{1,n} - X_{1,n} - b_1 X_{2,n} - b_2 X_{3,n} + F, (12)$$

where $c_1$, $c_2$, $b_1$, $b_2$ are parameters and the procedure for expressing the variables:

$$X_{1,n} = \sum_{i=-I_1}^{I_1} {}'\left(\left(\left(3I_1^2+3\right)/\left(2I_1^3+4I_1\right)\right)-\left(\left(2I_1^2+1\right)/\left(I_1^4+2I_1^2\right)\right)|i|\right)X_{tot,n+i}, \qquad (13)$$

$$X_{2,n} = \sum_{j=-I_2}^{I_2} {}'\left(\left(\left(3I_2^2+3\right)/\left(2I_2^3+4I_2\right)\right)-\left(\left(2I_2^2+1\right)/\left(I_2^4+2I_2^2\right)\right)|j|\right)\left(X_{tot,n+j}-X_{1,n+j}\right), \qquad (14)$$

$$X_{3,n} = X_{tot,n} - X_{2,n} - X_{1,n}, \qquad (15)$$

where $I_1$ and $I_2$ set the length of the intervals $[-I,I]$.

The parameters of any multi-level Lorenz's system (L96-2, L96-H, L05-2, L05-3) should be set so that all levels behave chaotically (the largest Lyapunov exponent of each level is positive) and that all levels have a significant difference in amplitudes and fluctuation rates. For the L-96 system (Eq. (1)), the chaotic behavior is determined by the value of $F$, and the number of variables $N$. Lorenz (2005) states that as long as $N \geq 12$ chaos is found, when $F > 5$ (for $N = 4$ it is when

$F > 12$ and for $N > 6$ when $F > 8$). In cases such as the L96-2 system ((3) and (4)), where the forcing $F$ acts only on the largest scale, the chaotic behavior of smaller scales is created by coupling. The size of the coupling is cascaded from the largest scale to the smaller ones. Because the values of the largest scale variables are determined by the forcing $F$, the $F$ value indirectly affects the smaller scales' chaotic behavior and must be chosen large enough to ensure chaotic behavior through coupling for all scales (levels). For the L05-2 system (Eq. (11)), variables are superposed features of a single set calculated by

Eqs (7) and (8). In addition to those mentioned above, this procedure affects the chaotic behavior, amplitude, and fluctuation rate of the levels, and the choice of $I$ between 10 and 20 may be optimal (Lorenz, 2005). In order to maintain the required properties of the two scales L05-2 system, Lorenz (2005) chose $N = 960$, $L = 32$, $I = 12$, $F = 15$, $b = 10$, and $c = 2.5$ (note that for L05-2 and L05-3 systems it is not possible to directly determine the amplitude and fluctuation rate of smaller scales using





spatiotemporal scaling factors $b$, because these values are mainly determined by the procedure for expressing variables and

the length of the intervals $[-I, I]$).

For the L05-3 system (Eqs. (12) – (15)), it is necessary to specify eight parameters. We tested that the values of coupling coefficients $c_1$ and $c_2$ do not affect the L05-3 system compared to the values of other parameters, and therefore for simplification $c_1 = 1$ and $c_2 = 1$. The parameter $F = 15$ is set the same as for other L05 systems. In order for the medium scale amplitude to be approximately ten times smaller than the large scale amplitude and the small scale amplitude to be

approximately ten times smaller than the medium scale amplitude and for the scales to have different oscillation rates (Fig. 1), the spatiotemporal scale factors are chosen $b_1 = 1$ and $b_2 = 10$ and interval lengths $I_1 = 20$, and $I_2 = 10$. $N = 390$ turned out to be most suitable for the chaotic behavior of all three levels. In the following, we will present numerical results of the L05-3 system (Eqs. (12) – (15)) with the parameters $N = 390$, $L = 13$, $J = 6$, $F = 15$, $b_1 = 1$, $b_2 = 10$, $c_1 = 1$, $c_2 = 1$, $I_1 = 20$, $I_2 = 10$, and calculated by a fourth-order Runge-Kutta method with a time step $\Delta t = 1/240$ or 0.5 hours. Initial conditions ($X_{tot,0,n}$

, $X_{1,0,n}$, $X_{2,0,n}$, $X_{3,0,n}$), which should be free of transient effect, are chosen as final values of arbitrary values integrated for 175200 steps or 10 years. Initial conditions and values of variables at days one, two, and three can be seen in Fig. 1.

## 3. Error growth in the L05-3 system

### 3.1. Numerical scheme for scale dependent error growth rates

By "error growth," we denote the growth of errors in the initial conditions, which limit predictability if a system is chaotic. A

numerical error growth experiment in a model system, therefore, consists of generating repeatedly a reference trajectory, which is considered to be the "truth" or verification, and a perturbed trajectory which is the numerical solution of the system for a perturbed initial condition. The time evolution of the difference vector between these two trajectories averaged over many repetitions then gives insight into the growth of prediction errors. In order for this scheme to be meaningful, we have to ensure that the reference trajectory is on the attractor of the system, that the repetition of this scheme samples the whole attractor with

correct weights (the invariant measure), and that initial perturbations point already into the locally most unstable direction since otherwise, errors might even shrink on short times (this is also a relevant issue in ensemble forecasts and there find its solutions in using bred vectors (Toth and Kalnay, 1997)). We solve these issues in the following way: We first integrate the system over 10 years, starting from arbitrary initial conditions, and assume that after discarding this transient, the trajectory is on the attractor. We continue to integrate this single trajectory and consider segments of it as reference trajectories for error

growth, i.e., the many reference trajectories are simply segments of one very long trajectory, which ensures not only that all these segments are located on the attractor, but that in addition, they sample the attractor according to the invariant measure. For the perturbed trajectories, we start with a random perturbation of the reference trajectory of very small amplitude and let this trajectory evolve over time before we start to determine its distance towards the reference trajectory. In other words, we





discard some initial time interval of error growth from our study since this is affected by some complicated transient behavior

before it starts to grow with the maximum Lyapunov exponent. However, due to the hierarchical nature of our model, the error

growth with the maximal Lyapunov exponent will saturate already for rather small error amplitudes and will be replaced by

slower error growth on larger scales, an effect which we will study in detail. The above described scheme was originally

introduced by Lorenz (1996).

In our system, the three spatial scales $X_1$, $X_2$, and $X_3$ cannot be separated in terms of a coordinate transform but are

intrinsically coupled and superimposed in the variables $X_{tot}$ of the system. The initial conditions of the "reality" are called

$X_{tot,0,n}$, from which one finds $X_{1,0,n}$, $X_{2,0,n}$, and $X_{3,0,n}$ through Eqs. (13), (14) and (15). The initial values of the "prediction"

are then $X'_{tot,0,n} = X_{1,0,n} + X_{2,0,n} + X_{3,0,n} + e_{3,n} = X_{tot,0,n} + e_{3,n}$, where $e_{3,n}(0)$ are the initial errors randomly selected from the

normal distribution $ND(\mu = 0; \sigma = 0.01)$. Hence, these initial conditions differ only by the small perturbation $e_{3,n}(0)$. Since

the state of the model $X_{tot}$ is the sum over the three components, any arbitrary but small error with spatially uncorrelated

components affects only $X_{3,0,n}$. Only a spatially correlated initial error would appear in another component, but since this error

would immediately propagate into the small scale variables and then grow fastest in these, a perturbation with initial errors in

$X_{3,0,n}$ is the only practical choice. From $X_{tot,0,n}$ and $X'_{tot,0,n}$ Eqs. (12) are integrated forward for 41.7 days ($K = 2000$ steps).

In each time step $k$ of the numerical integration, $X_{\tau,k,n}$, and $X'_{\tau,k,n}$ are obtained. The size of the error at a given time $k\Delta t$ is

$e_{\tau,n}(k \cdot \Delta t) = X'_{\tau,k,n} - X_{\tau,k,n}$, where $k = 1,\ldots,K$, $n = 1,\ldots,N$ and $\tau = tot,1,2,3$. We perform $M = 400$ runs in order to

calculate the average error growth. In each new run, the initial values $X_{\tau,0,n}$ are the last values $X_{\tau,K,n}$ of the previous run. The

average initial error growth $E(t)$ is defined as the geometric mean of the runs of the Euclidean distances between "reality"

and "prediction":

$$E_\tau(k \cdot \Delta t) = \sqrt[2M]{\prod_{m=1}^{M}\left(\frac{1}{N}\sum_{n=1}^{N}e_{\tau,n,m}^2(k \cdot \Delta t)\right)}, \quad (16)$$

where $\tau = tot,1,2,3$. The initial transient behavior (0.7 days) of the average error growth $E(t)$ is discarded. We do this since

the random initial errors perturb the initial state off the attractor. On very short times, the perturbed trajectory relaxes back to

the attractor (Brisch and Kantz, 2019; Bednar et al., 2014) so that the error does not grow. Only after this transient one observes

the characteristic error growth of the system. In real weather forecasts, the data assimilation scheme 4D-VAR usually ensures

that the (error prone) analysis which is used as the initial condition of the forecast is on the attractor, and also in ensemble

forecasts, the perturbations to the anlysis are, e.g., by the usage of bred vectors, done in a way that all ensemble members start

on (or at least very close to) the attractor. As a result, we have numerical averages for the error growth as a function of time





steps after perturbing the reference trajectories in the full phase space and for $\tau = 1, 2, 3$. We can convert these results into the error growth rate as a function of time, and into the error growth rate as a function of the error magnitude.

### 3.2. Functional forms for error growth rates

Let us first consider a classical low dimensional chaotic system. If the initial errors were infinitesimal, one could follow their

growth for infinite times and define the maximal Lyapunov exponent of the system as $\Lambda_{max} = \lim_{t \to \infty} \lim_{\epsilon \to 0} \frac{1}{t} \ln |X'_\tau(t) - X_\tau(t)|$

, with a time and error independent growth rate $\Lambda$. This exponential growth is associated with single scale systems, infinitesimal initial error, and the early part of the error growth (Bednar et al., 2014). In reality, since the initial perturbation is non-infinitesimal, the exponential error growth will cross over to a constant: the distance between the "true" and the perturbed trajectories cannot become larger than the largest distance on the attractor. Hence, for large times. the average "error" is then

the average distance of two arbitrary points on the attractor, averaged over the invariant measure. In order to take this saturation effect into account, we use the extended exponential law for the error growth rate: $\lambda(E) = \Lambda(1 - E/E_\infty)$, where $E_\infty$ is the saturation value of the respective error. Notice that this extension does not involve any free parameter other than the measurable saturation value $E_\infty$.

We calculate the error growth rate in our L05-3 system using the method of Sprott (2006) ($e_{\tau,n}(0) = 10^{-10}$, $k = 10$ and $k = 100$

, $M = 10^5$). In the following, $E_\tau(t)$ denotes mean over many initial conditions of the Euclidian distances between the "true" and the perturbed trajectories at time $t$ after initializing the perturbation, measured in the subspace of the coordinates $\tau$, where $\tau \in \{tot, 1, 2, 3\}$. Numerically, we find $E_{tot,\infty} = 7.4$, $E_{1,\infty} = 6.6$, $E_{2,\infty} = 1.4$, $E_{3,\infty} = 0.3$ (units). We determine the maximal Lyapunov exponents in all 4 cases and find the values $\Lambda_{theor} = \Lambda_{tot,theor} = \Lambda_{3,theor} \approx 2.5$ $(days)^{-1}$ and $\Lambda_{1,theor} = \Lambda_{2,theor} \approx 2$ $(days)^{-1}$. The similarity of the values $\Lambda_{tau,theor}$ for all levels $\tau = 1, 2, 3$ indicates that they are coupled, so

that the maximal Lyapunov exponent when calculated in the double limit $E_0 \to 0$ and $t \to \infty$ shows up in arbitrary subsystems. These values must not be confused with the Lyapunov exponents characterizing the different levels which we will calculate later: The evolution of the errors $E_\tau$ can always be studied in a way to see the largest exponent of the system (done here), but also in a way to see a value which would be the exponent of the corresponding sub-sytem if one were able to isolate this. In the context of the coupled system, these sub-system exponents as just some other positive exponents of the full system,

since Eq. (12) in total has, as every dynamical system, as many Lyapunov exponents as it has degrees of freedom (here: $N$ =390), were several of these can be positive.





The L05-3 system is designed with three spatial and temporal scales, so the error growth rate $\lambda(E) \approx \dfrac{1}{\Delta t} \ln(E(t + \Delta t) / E(t))$ is expected to be a function of the error magnitude $E$: after the small scale errors are no longer growing, the large scales errors continue to grow at a lower rate. Brisch and Kantz (2019) described this dependence by a power law:

$$\lambda_p(E) := \frac{d \ln(E)}{dt} = \frac{\dot{E}}{E} = aE^{-\sigma},$$ (17)

with an exponent $\sigma$ and a coefficient $a > 0$. By integrating Eq. (17) over time, the power law dependence of the error growth rate on the error magnitude translates into a power law growth of errors over time:

$$E_p(t) = \left( E_0^\sigma + a\sigma t \right)^{1/\sigma}.$$ (18)

Similar to the exponential law, this power law does not take the saturation of errors at their largest scale into account. We, therefore, multiply the right hand side by $(1 - E / E_\infty)$ amd arrive at the extended power law:

$$\lambda_w(e) := \frac{d \ln E}{dE} = \frac{\dot{E}}{E} = aE^{-\sigma} \left( 1 - \frac{E}{E_\infty} \right).$$ (19)

Unfortunately, the time integration in order to arrive at an expression of $E_w(t)$ cannot be done analytically but numerically once the parameters are fixed.

A different description of scale dependent error growth rate was proposed by Zhang et al. (2019), namely a quadratic model which we write down directly in its extended form containing the saturation effect:

$$\lambda_q(E) := \frac{d \ln E}{dt} = \frac{\dot{E}}{E} = \alpha + \beta E^{-1} \left( 1 - \frac{E}{E_\infty} \right),$$ (20)

where $\alpha$ is a synoptic scale error growth rate and $\beta$ is an upscale error growth rate from small scale processes, which was originally designed to describe model error (Savijarvi, 1995; Dalcher and Kalney, 1987). By integrating Eq. (20) over time, we find

$$E_q(t) = E_\infty - \frac{\left( E_\infty - \beta / \alpha \right)}{1 - \left( E_0 + \beta / \alpha \right) \exp\left[ (\alpha + \beta / E_\infty)t \right] / \left( E_0 - E_\infty \right)}.$$ (21)

When removing the saturation factor $(1 - E / E_\infty)$ in Eq. (21), then it is what was called the quadratic model,

$$\lambda_r(E) := \frac{d \ln E}{dt} = \frac{\dot{E}}{E} = \alpha + \beta E^{-1},$$ (22)









with the time evolution of the error

$$E_r(t) = E_0 + \left( E_0 + \frac{\beta}{\alpha} \right)\left( \exp[\alpha t] - 1 \right),\tag{23}$$

which is suitable for the first few days of the error growth.

To summarize, we can test the validity of the following laws for scale dependent error growth rates and for the error growth over time: a constant Lyapunov exponent and hence an exponential error growth, as it is expected for the initial time of very small initial errors in a low dimensional chaotic system; the extension of this behavior with a saturation factor $(1 - E / E_\infty)$ expected to be valid for all times in a low dimensional chaotic system; the (extended) quadratic law proposed in (Zhang et al.

2019), and the (extended) power law growth proposed in Brisch and Kantz (2019).

In Figs. 2, 3 and 4, we present the numerical results of the error growth over time of the errors $E_\tau(t)$ for $\tau \in \{tot, 1, 2, 3\}$, and the corresponding error growth rates as a function of the error magnitude.

The power law $\lambda_p(E)$ (Eq. (17)) approximates the L05-3 system error growth rate $\lambda_{tot}(E)$ in the interval $E_{tot}(t) \in \left[ E_{tot,0}, 1.5 \right]$ (Fig. 2a). For larger errors, there is no "next level" of the hierarchy any more, so that the power law evidently cannot be valid.

For $E_{tot}(t) \in \left[ 1.5, \ E_{tot,\infty} \right]$, the empirical error growth rate $\lambda_{tot}(E)$ (Fig. 2a) decreases much faster than the power $\lambda_p(E)$, due to saturation of the errors at $E_\infty$. Hence, the power law $E_p(t)$ (Eq. (18)) yields a good approximation of L05-3 system error growth $E_{tot}(t)$ only in the early part of the growth within six days (Fig. 2b), where we find the numerical values $\sigma = 0.5$ and $a = 0.41$ (units$^{0.5}$/day). This power law needs to be corrected on times beyond 6 days in order to be able to model the saturation of errors, called the extended power law above. Inserting the numerically observed saturation value $E_\infty = 7.4$, a fit of $\lambda_w(E)$

Eq. (19) yields $\sigma = 0.47$, $a = 0.46$ (units$^{0.47}$/day). With this extension, we are able to suitably approximate the L05-3 system error growth $E_{tot}(t) \in \left[ E_{tot,0}, \ E_{tot,\infty} \right]$ (Fig. 2a) on the entire range of errors. The numerical time integration of Eq. (19) correspondingly well approximates the L05-3 system error growth $E_{tot}(t)$ from the initial conditions $E_0$ to the limit (saturated) value $E_{tot,\infty}$ (Fig. 2b).

We also fit the quadratic model $\lambda_r(E)$, Eq. (22), to the numerically obtained L05-3 system error growth rate $\lambda_{tot}(E)$. It can

reproduce the error growth rate in the interval $E_{tot}(t) \in \left[ E_0, \ 1.5 \right]$ as well, but not as accurately as the power law (Fig. 3a). For larger errors In the interval $E_{tot}(t) \in \left[ 1.5, \ E_{tot,\infty} \right]$, the quadratic model $\lambda_r(E)$ decreases more slowly than the L05-3 system error growth rate $\lambda_{tot}(E)$ and the power law $\lambda_p(E)$ (Fig. 3a and 2a). The solution of the quadratic model for the time evolution of the error, $E_r(t)$ (Eq. (23)), with $\alpha = 0.25$ (day$^{-1}$) and $\beta = 0.13$ (units/day) determined from the approximation is similar to $E_p(t)$ with a given coefficient and exponent, but it does not approximate the data as accurately (Fig. 3b). The





factor $(1-E_\infty)$ with $E_\infty = 7.3$ in the extended quadratic model, Eq. (20), yields a much better approximation along the entire

time interval, $E_{tot}(t) \in [E_{tot,0}, E_{tot,\infty}]$ (Fig. 3a), but it is slightly less accurate than the extended power law in the interval

$E_{tot}(t) \in [E_{tot,0}, 1.5]$. This inaccuracy is significant for the solution of the extended quadratic model $E_q(t)$ (Eq.(21)) with

$\alpha = 0.2$ (day$^{-1}$) and $\beta = 0.19$ (units/day) determined from the approximation, where the early part of growth is the least

similar to error growth $E_{tot}(t)$ (Fig. 3b).

### 3.3. Self-consistent explanation of the observed approximate power law


Britch and Kantz (2019) derived the value of the exponent $\sigma$ of the power law Eq. (17) from other system properties. They

argue that the power law is a result of the superposition of the error growth on different spatial scales with different growth

rates. Translated into our L05-3 system it should work as follows: The small scale error growth $E_3(t)$ should follow the

(extended) exponential growth (solution of $\dot{E}/E = \Lambda(1-E/E_\infty)$), with $\Lambda_3$ and a saturation scale $E_{3,\infty}$. After its saturation, the

medium scale error continues to grow, but with a lower rate $\Lambda_2$, and will saturate at a larger scale $E_{2,\infty}$. Beyond that, the total

error growth can only be driven by the growth of the large scale errors with an even slower growth rate $\Lambda_1$ and saturation scale

$E_{1,\infty}$. If the model is designed such that $\Lambda_2 = c\Lambda_1$ and $\Lambda_3 = c^2\Lambda_1$, while $E_{2,\infty} = E_{1,\infty}/b$ and $E_{3,\infty} = E_{1,\infty}/b^2$, then it was

argued that the exponent of the power law should be $\sigma = \ln c / \ln b$.

The model of Britch and Kantz (2019) was constructed of weakly coupled identical sub-systems, with additional scaling

parameters for the amplitude and time of the different subsystems. Therefore, there were rather well defined error growth rates

$\Lambda_\tau$, $\tau = 1,2,3$, and it was easily possible to tune these so that $c^2\Lambda_1 = c\Lambda_2 = \Lambda_3$ and $E_{3,\infty} = E_{2,\infty}/b = E_{1,\infty}/b^2$. In our L05-3

model, all this is not the case. The advantage of our model, however, is that it is much closer to the phenomenology of an

atmospheric model: In a single field $X_{tot}$, we observe both large scale and small scale structures, and we observe both fast and

slow motion.

In the L05-3 model, we have determined the error growth of a given level by the study of the distance between reference

trajectories and perturbed trajectories in the corresponding coordinates $X_\tau$, $\tau = 1,2,3$, see Fig. 4. It is straight-forward to

identify the saturation values $E_{\tau,\infty}$. We describe here how we find approximate values for the growth rates $\Lambda_\tau$ in the extended

exponential law ($\lambda(E) = \Lambda(1-E/E_\infty)$) from our numerics. We can then determine $c$ and $b$ and hence a theoretical value for

$\sigma$, which we will compare to the numerical fit.

By Sprott's method (Sprott 2006) we calculate the error growth rates of the three levels separately and the total error growth

rate. As one can see in Fig. 4, $\lambda_\tau(E)$ can be well described by the extended power law $\lambda_w(E)$ on the whole range of $E$ for

all $\tau$. The individual contributions of the levels (scales) $X_\tau$, $\tau = 1,2,3$ to the error growth rates $\lambda_\tau(E)$ are then identified as





parts that fulfill the extended exponential growth $\lambda_e(E)$. Numerically, we identify the following values: The extended growth rates are $\Lambda_3 = 3.6$ (day$^{-1}$), $\Lambda_2 = 0.8$ (day$^{-1}$), and $\Lambda_1 = 0.28$ (day$^{-1}$). The corresponding saturation scales are determined as

$E_{3,\infty_e} = 0.06$ (units), $E_{2,\infty_e} = 0.55$ (units), and $E_{1,\infty_e} = 6.1$ (units). The values for levels $\tau = 2$ are extracted from the range where the effect of $\lambda_3(E)$ is no longer present (Fig 4b), while for $\tau = 1$ we consider the late stage of the error growth rate $\lambda_1(E)$ (Fig 4c). Following the arguments from above, we derive $\sigma$ as $\sigma = \ln b_1 / \ln c_1 = 0.47$ where $b_1 = E_{2,\infty_e} / E_{3,\infty_e}$ and $c_1 = \Lambda_{2,e} / \Lambda_{1,e}$. We get the same value of $\sigma = \ln b_2 / \ln c_2 = 0.47$ where $b_2 = E_{1,\infty_e} / E_{2,\infty_e}$, and $c_2 = \Lambda_{3,teor} / \Lambda_{2,e}$, even though that $b_1 \neq b_2$ and $c_1 \neq c_2$. Note that $c_2$ uses $\Lambda_{3,teor}$ instead of $\Lambda_3$. This is because $\Lambda_3$ should be computed from the ranges

where $E$ and $t$ are small/short but which include some transient behavior (Fig. 4a), and therefore $\Lambda_{3,teor}$ is a more appropriate value.

These derived values for $\sigma$ are in perfect agreement with the empirical fit of the extended power law $\lambda_w(E)$, which yields $\sigma = 0.47$, and it is close to the value $\sigma = 0.50$ of the power law $\lambda_p(E)$ without the saturation effect.

The coupling of the levels has the consequence that one cannot define the Lyapunov exponents for the individual levels in a

mathematically rigorous way. We calculated them as $\Lambda_1(E_{2,\infty_e}) = 0.28 = const.$ (day$^{-1}$) at $E_{2,\infty_e} = 0.55$ and $\Lambda_2(E_{3,\infty_e}) = 0.8 = const.$ (day$^{-1}$) at $E_{3,\infty_e} = 0.06$ using the extended exponential law. It is well justified to use these to determine the exponent $\sigma$. However, they do not fit the error growth rate $\lambda_{tot}(E)$ ($\lambda_{tot}(E_{2,\infty_e}) \neq \Lambda_1$, $\lambda_{tot}(E_{3,\infty_e}) \neq \Lambda_2$, Fig. 4d). Instead, a fit to $\lambda_{tot}(E)$ yields $\lambda_{tot}(E_{2,\infty}) = \Lambda_1 = 0.28$ (day$^{-1}$) in the limit value of medium scale error growth $E_{2,\infty} = 1.4$ and $\lambda_{tot}(E_{3,\infty}) = \Lambda_2 = 0.8$ (day$^{-1}$) in the limit value of small scale error growth $E_{3,\infty} = 0.3$ (Fig. 4d). If we approximate the

values $\Lambda_1(E_{2,\infty_e})$ and $\Lambda_2(E_{3,\infty_e})$ by the power law $\lambda_p(E)$ with the exponent $\sigma = 0.5$, we get the coefficient $a = 0.2$ (units$^{0.5}$/day), and by the power law $\lambda_p(E)$ with the exponent $\sigma = 0.47$, we get $a = 0.21$ (units$^{0.47}$/day). These power laws should describe the error growth of the system without coupling (the extended power law was not used because $(1 - E/E_\infty)$ is negligible in this area) and if we compare these values with the values of the power laws ($\lambda_{tot,p}(E)$: $\sigma = 0.5$, $a = 0.41$ (units$^{0.5}$/day), $\lambda_{tot,w}(E)$: $\sigma = 0.47$, $a = 0.46$ (units$^{0.46}$/day)) that approximate the error growth rate of the L05-3 system $\lambda_{tot}(E)$

(the system with coupling), we find that the value of the coefficient $a$ changed. We, therefore, conclude that the coefficient $a$ is subject to the degree of coupling of the system.





Values $\alpha = 0.25$ (day$^{-1}$) for the quadratic model $\lambda_r(E)$ (Eq. (22)) and $\alpha = 0.2$ (day$^{-1}$) for the extended quadratic model $\lambda_q(E)$ (Eq. (20)) should describe synoptic scale error growth rate (Zhang et al., 2019). For the L05-3 system, it is $\Lambda_1 = 0.28$ (day$^{-1}$) obtained from the approximation by the extended exponential growth ($\lambda(E) = \Lambda(1 - E/E_\infty)$) for $X_1$. We can see that

$\Lambda_1$ is closer to $\alpha$ from the quadratic model $\lambda_r(E)$, but none of $\alpha$ describes $\Lambda_1$ exactly. Values $\beta = 0.13$ (units/day) and $\beta = 0.19$ (units/day) that by Zhang et al. (2019) should describe upscale error growth rate from small scale processes could not be identified or described in the L05-3 system.

## 4.  Error growth in the ECMWF forecast system

The error growth $E_{EFS}(t)$ of the ECMWF forecasting system's 500 hPa geopotential height values (Magnusson, 2018) is

calculated (Magnusson and Kallen, 2013) as annual averages over the Northern Hemisphere ($20°$–$90°$) obtained daily from 1 January 1986 to 31 December 2011. As "errors" one uses the differences between two operational forecasts issued with one day lag for the same day, in order to eliminate the effects of model errors. Specifically, we evaluete these for 27 different lead times and used the following time intervals in hours: 0–24, 6–30, 12–36, 18–42, 24–48, 30–54, 36–60, 42–66, 48-72, 54–78, 60–84, 66–90, 72–96, 78–102, 84–108, 90–114, 96–120, 108–132, 120–144, 132–156, 144–168, 156–180, 168–192, 180–204,

192–216, 204–228, 216–240. Detailed information about calculating the error growth of the ECMWF forecasting system can be found in Lorenz (1982). The error growth rate $\lambda_{EFS}(E) := d\ln E/dt \approx \ln(E(t+\Delta t)/E(t))/\Delta t$ with $\Delta t = 6$ hours for the first seventeen time steps and $\Delta t = 12$ hours for the remaining ten time steps and the error growth $E_{EFS}(t)$ are calculated between the first and tenth day. The differences 0–24, 6–30, 12–36 are discarded because $\lambda_{EFS}(E)$ for these differences is either increasing or constant due to transient behavior, see Fig. 5.

Despite these missing parts of the error growth $E_{EFS}(t)$, Bednar et al.(2020) showed that the extended quadratic model $E_q(t)$ Eq. (21) is more accurate than the extended exponential growth (solution of $\dot{E}/E = \Lambda(1 - E/E_\infty)$). Because $E_{EFS}(t)$ calculated in the above mentioned way should be insensitive to any model error which therefore can not affect the error growth, the extended quadratic model parameter $\beta$ describes the effect of small scale processes, which justifies to consider the error growth rate $\lambda_{EFS}(E)$ for all annual averages as scale-dependent and to approximate it also by the extended power law $\lambda_w(E)$

Eq. (19). The quadratic model $\lambda_r(E)$ (Eq. (22)) and the power law $\lambda_p(E)$ (Eq. (17)) cannot be used due to the lack of an early part of the error growth $E_{EFS}(t)$. The extended power law $\lambda_w(E)$ (Eq. (19)) matches well with the annual averages of the error growth rate $\lambda_{EFS}(E)$ of the ECMWF forecasting system (Fig. 5) with the values of the exponent $\sigma_{EFS}$ displayed in Fig. 6b ($\bar{\sigma}_{EFS} = 0.21 \pm 0.07$), the value of the coefficient a $_{EFS}$ shown in Fig. 6a ($\bar{a}_{EFS} = 0.93 \pm 0.18$ m$^\sigma$/day) and with the limit


(saturated) values $E_{EFS,\infty_w}$ displayed in Fig. 8 ($\bar{E}_{EFS,\infty_w} = 114 \pm 7$ m). Exponents $\sigma_{EFS}$, coefficients $a_{EFS}$, and limit values

$E_{EFS,\infty_w}$ do not have significant trends over the years (Fig. 6 and 8). The extended quadratic model $\lambda_q(E)$ Eq. (20)

approximates annual averages of the error growth rate $\lambda_{EFS}(E)$ of the ECMWF forecasting system (Fig. 5) with the values of

the parameter $\alpha_{EFS}$ shown in Fig. 7a ($\bar{\alpha}_{EFS} = 0.35 \pm 0.04$ day$^{-1}$), the value of the parameter $\beta_{EFS}$ displayed in Fig. 7b (

$\bar{\beta}_{EFS} = 2.8 \pm 0.9$ m/day) and with the limit (saturated) values $E_{EFS,\infty_q}$ displayed in Fig. 8 ($\bar{E}_{EFS,\infty_q} = 111 \pm 7$ m). Again,

parameters $\alpha_{EFS}$, $\beta_{EFS}$, and limit values $E_{EFS,\infty_q}$ do not have significant trends over the years (Fig. 7 and 8). The

approximations by the extended quadratic model $\lambda_q(E)$ and the extended power law $\lambda_w(E)$ differ in parts where data are

not available (Fig. 5). The limit values $E_{EFS,\infty_w}$ and initial errors $E_{EFS,0_w}$ of the extended power law are greater than the limit

values $E_{EFS,\infty_q}$ and initial errors $E_{EFS,0_q}$ of the extended quadratic model (Fig. 5 and 8). If we set the initial errors of both

approximations to the same value, the error growth of the extended quadratic model $E_{EFS,q}(t)$ approximation would grow

faster and would reach the limit value $E_{EFS,\infty}$ earlier than the error growth of the power law $E_{EFS,w}(t)$. This difference is more

significant when $E_0 \to 0$. Of particular interest is the limit of predictability. We, therefore, study the time when the error $E(t)$

reaches 95% of the limit error $E_\infty$ for the size of the initial error $E_0 \approx 3$ m, which is an accepted value for current global

operational NWP models (Zhang et al., 2019). The limit time is 14 days for the extended quadratic model $\bar{E}_{EFS,q}(t)$ when using

the above mentioned parameter values for $\bar{\alpha}_{EFS}$, $\bar{\beta}_{EFS}$, and $\bar{E}_{EFS,\infty_q}$ (Fig. 9a). For the extended power law $\bar{E}_{EFS,w}(t)$, using

the above listed values for the exponent $\bar{\sigma}_{EFS}$, coefficient $\bar{a}_{EFS}$, and limit values $\bar{E}_{EFS,\infty_w}$, it is 15 days (Fig. 9b). When the size

of the initial error $E_0$ is reduced to $E_0 \approx 0.1$ m which is realistic for the current global experimental NWP models (Zhang et

al., 2019), this limit time is 15 days for the extended quadratic model $\bar{E}_{EFS,q}(t)$ (Fig. 9a), and 18 days for the extended power

law $\bar{E}_{EFS,w}(t)$ (Fig. 9b). Both models have an intrinsic predictability limit even if the size of the initial error $E_0 \to 0$, which

is 15 dats for the extended quadratic model (Fig. 9a) and 22 days for the extended power law $\bar{E}_{EFS,w}(t)$ (Fig. 9b).

## 5. Conclusion and discussion

We designed a three levels (scales) system (L05-3, Eqs. (12) – (15)) with the parameters $N = 390$, $L = 13$, $J = 6$, $F = 15$,

$b_1 = 1$, $b_2 = 10$, $c_1 = 1$, $c_2 = 1$, $I_1 = 20$, $I_2 = 10$. These parameters were chosen in such a way that all levels behave

chaotically, i.e., the largest Lyapunov exponents of each level is positive, and that all levels have a significant difference in

amplitudes and fluctuation rates (Fig. 1). In this system, the error growth rates $\lambda_\tau(E)$ and the error growths as a function of



time $E_\tau(t)$ were calculated for the system as a whole ($\tau = tot$) and for the different levels ($\tau = 1, 2, 3$). We fitted the

numerically obtained results by the power law $\lambda_p(E)$ (Eq. (17)), the extended power law $\lambda_w(E)$ (Eq. (19)), the quadratic

hypothesis $\lambda_r(E)$ (Eq. (22)), the extended quadratic hypothesis $\lambda_q(E)$ (Eq. (20)) and their corresponding time integrations

$E(t)$. Without the saturation terms $(1 - E/E_\infty)$ both the power law and the quadratic hypothesis fail to provide good fits for

larger errors or longer times (Fig. 2 and 3) but are reasonable for the early parts of error growth when the initial error is small.

The quadratic hypothesis does not provide an as good fit as the extended power law does. Its parameter $\beta$ that by Zhang et

al. (2019) should describe the upscale error growth rate from small scale processes could not be identified or described in the

L05-3 system. In contrast, the extended power law $\lambda_w(E)$ and $E_w(t)$ with the exponent $\sigma = 0.47$, the coefficient $a = 0.46$

(units $^{0.47}$/day), and $E_\infty = 7.36$ best describes the error growth rate $\lambda_{tot}(E)$ and the error growth $E_{tot}(t)$ of the L05-3 system

on the whole range of times and error magnitudes. Britch's and Kantz's (2019) definition of the exponent $\sigma$ was confirmed

and extended to cases when $c_1 \neq c_2$ ($c$ is the ratio of the rapidness of a smaller scale compared to the rapidness of a larger

scale) and $b_1 \neq b_2$ ($b$ is the ratio of a smaller scale amplitude compared to a larger scale amplitude) for

$\sigma = \ln c_1 / \ln b_1 = \ln c_2 / \ln b_2$. It was shown that the coefficient $a$ determines the degree of the system's coupling because the

same value of the exponent $\sigma$ is valid for the same system with and without coupling.

We also checked the appropriateness of the extended quadratic hypothesis and the extended power law to describe the error

growth rate $\lambda_{EFS}(E)$ and the error growth over time $E_{EFS}(t)$ of the ECMWF forecasting system's 500 hPa geopotential height

values over the 1986 to 2011 period. Their behaviors differ in parts where data are not available (Fig. 5), while they agree

quite well and both describe the numerical observations on the main part of the data. Therefore, it is not possible to assess

which approximation is the more appropiate one for the description of the entire length of $\lambda_{EFS}(E)$ and $E_{EFS}(t)$ from the initial

to the limit (saturated) error. One might argue, however, that the study of the L05-3 system whose behavior is best described

by the extended power law $\lambda_w(E)$, where it is theoretically substantiated and because we can observe the similarities of the

differences between the data and the approximations in both systems ($\bar{E}_{EFS,\infty_{teor}} > \bar{E}_{EFS,\infty_w} > \bar{E}_{EFS,\infty_q}$ (Fig. 8) and

$E_{tot,\infty} > E_{tot,\infty_w} > E_{tot,\infty_q}$ ; $\bar{\alpha}_{teor,w} > \bar{\alpha}_{EFS,w}$ (Fig. 7) and $\Lambda_1 > \alpha_{tot,w}$), that the extended power law $\lambda_w(E)$ is also a valid

description of the ECMWF forecasting system. From the average of the fit parameters over many years we calculate that the

intrinsic limit to predictability is 22 days in the idealized case of perfect initial conditions, which is in nice agreement with

Krishnamurthy (2019).



**Code and data availability**

The ECMWF forecasting system dataset was obtained from the personal repository of Linus Magnusson (Magnusson, 2013). L05-3 system dataset, products from the ECMWF forecasting system dataset, codes, and figures were conducted in Wolfram Mathematica, and they are permanently stored at http://www.doi.org/10.17605/OSF.IO/2GC9J.

**Author contributions**

H.B. proposed the idea, carried out the experiments, and wrote the paper. H.K. supervised the study and co-authored the paper.

**Competing interests**

The authors declare that they have no conflict of interest.

**Acknowledgements**

The authors are grateful to Linus Magnusson for offering Dataset (ECMWF forecasting system) from his personal repository.

**Financial support**

This study was supported by the Czech Science Foundation, through grant 19-16066S.

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





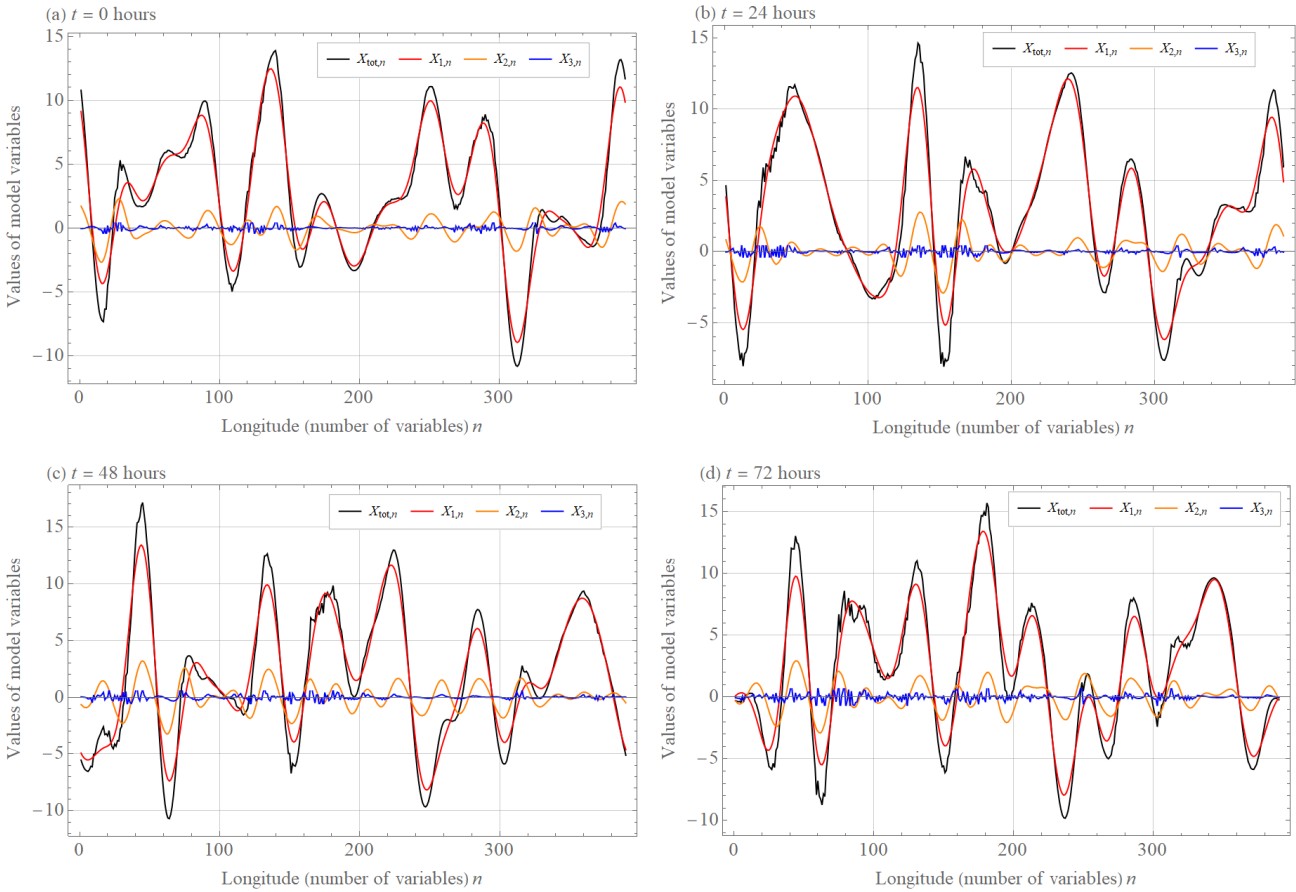

**Figure 1(a) Initial conditions (after discrding a transient of 10 years) and values of variables ($X_{tot}$, $X_1$, $X_2$, $X_3$) at days (b) one, (c) two, and (d) three of the L05-3 system (Eqs. (13) – (15)) with the parameters $N = 390$, $L = 13$, $J = 6$, $F = 15$, $b_1 = 1$, $b_2 = 10$, $c_1 = 1$, $c_2 = 1$, $I_1 = 20$, $I_2 = 10$ calculated by a fourth-order Runge-Kutta method with a time step $\Delta t = 1/240$ or 0.5 hours.**



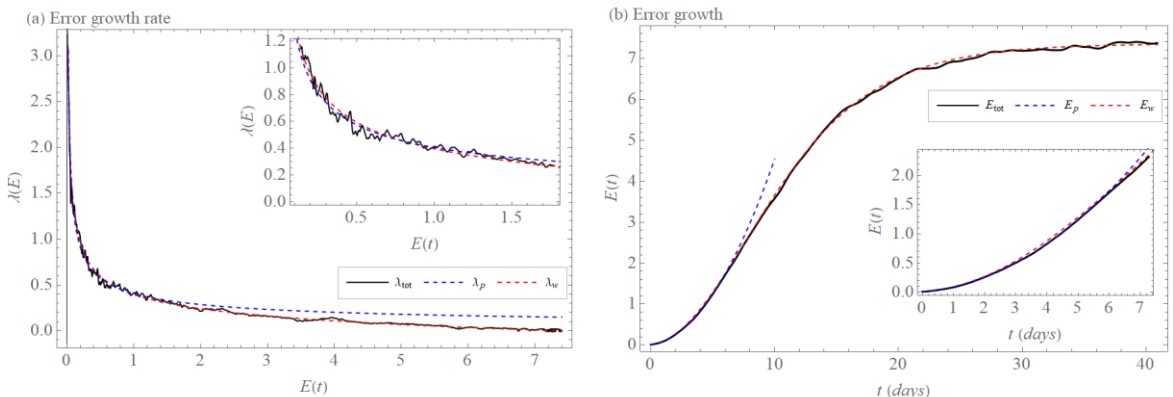

**Figure 2. (a) Error growth rate** $\lambda_{tot}(E)$ **of the L05-3 system (black). Power law** $\lambda_p(E)$ **(Eq. (19)) with the exponent** $\sigma = 0.5$ **and the**
**coefficient** $a = 0.41$ **(units$^{0.5}$/day) (** $\lambda_{tot,p}(E)$ **, blue, dashed) and extended power law** $\lambda_w(E)$ **(Eq. (26)) with the exponent** $\sigma = 0.47$ **,**
**the coefficient** $a = 0.46$ **(units$^{0.47}$/day), and** $E_\infty = 7.36$ **(units) (** $\lambda_{tot,w}(E)$ **, red, dashed) that approximate** $\lambda_{tot}(E)$ **. (b) Error growth**
$E_{tot}(t)$ **(Eq. (17)) (black), solution** $E_{tot,p}(t)$ **(blue, dashed) of** $\lambda_{tot,p}(E)$ **(Eq. (19)), and numerical solution of** $\lambda_{tot,w}(E)$ **(red, dashed).**

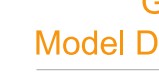
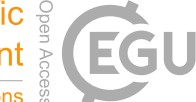

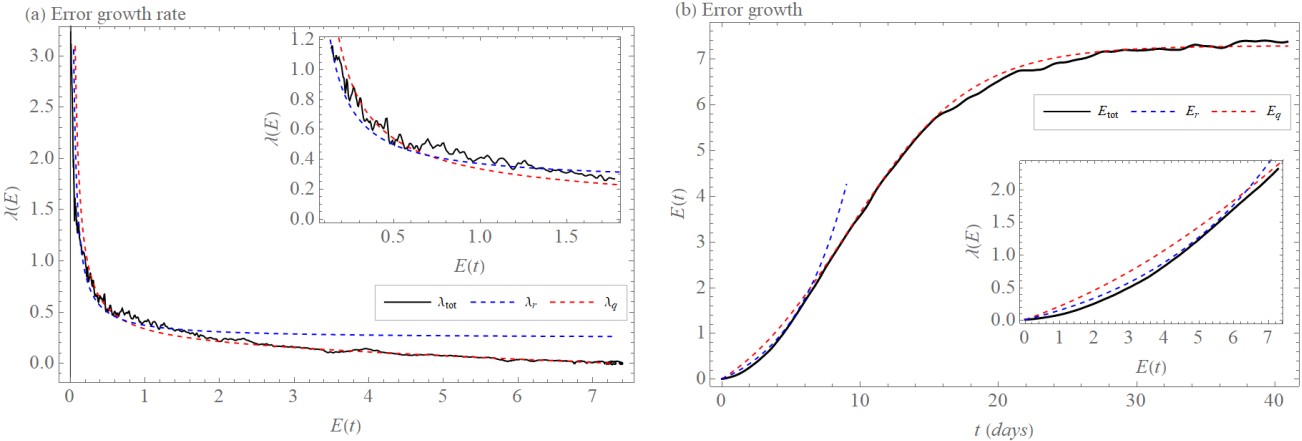

**Figure 3. (a) Error growth rate** $\lambda_{tot}(E)$ **of the L05-3 system (black). Quadratic model** $\lambda_r(E)$ **(Eq. (22)) with** $\alpha = 0.25$ **(day⁻¹) and**

$\beta = 0.13$ **(units/day) (** $\lambda_{tot,p}(E)$ **, blue, dashed) and extended quadratic model** $\lambda_q(E)$ **(Eq. (20)) with** $\alpha = 0.2$ **(day⁻¹), the coefficient** $\beta = 0.19$ **(units/day), and** $E_\infty = 7.3$ **(units) (** $\lambda_{q,w}(E)$ **, red, dashed) that approximate** $\lambda_{tot}(E)$ **. (b) Error growth** $E_{tot}(t)$ **(Eq. (17)) (black), solution** $E_{tot,r}(t)$ **(blue, dashed) of** $\lambda_{tot,r}(E)$ **(Eq. (22)), and solution** $E_{tot,q}(t)$ **(blue, dashed) of** $\lambda_{tot,q}(E)$ **(Eq. (21)) (red, dashed).**





**Figure 4. (a) Error growth rate** $\lambda_3(E)$ **of the L05-3 system's** $X_3$ **scale (black). Extended exponential growth** $\lambda_e(E)$ **(Eq. (24)) with** $\Lambda_3 = 3.6$ **(day$^{-1}$) and** $E_{3,\infty_e} = 0.06$ **(units) (** $\lambda_{3,e}(E)$ **, red, dashed) and extended power law** $\lambda_w(E)$ **(Eq. (26)) with the exponent** $\sigma = 0.75$**, the coefficient** $a = 0.13$ **(units$^{0.75}$/day), and** $E_{3,\infty_w} = 0.3$ **(units) (** $\lambda_{3,w}(E)$ **, blue, dashed) and** $\Lambda_{teor} = \Lambda_{tot,teor} = \Lambda_{3,teor} \approx 2.5$ **(day$^{-1}$) (Sprott's (2006) method, orange, dotted). (b) Error growth rate** $\lambda_2(E)$ **of the L05-3 system's** $X_2$ **scale (black). Extended exponential growth** $\lambda_e(E)$ **(Eq. (24)) with** $\Lambda_2 = 0.8$ **(day$^{-1}$) and** $E_{2,\infty_e} = 0.55$ **(units) (** $\lambda_{2,e}(E)$ **, red, dashed) and extended power law** $\lambda_w(E)$ **(Eq. (26)) with the exponent** $\sigma = 0.44$**, the coefficient** $a = 0.25$ **(units$^{0.44}$/day), and** $E_{2,\infty_w} = 1.4$ **(units) (** $\lambda_{2,w}(E)$ **, blue, dashed) and error growth rate** $\lambda_3(E)$ **of the L05-3 system's** $X_3$ **scale (orange). (c) Error growth rate** $\lambda_1(E)$ **of the L05-3 system's** $X_1$ **scale (black). Extended exponential growth** $\lambda_e(E)$ **(Eq. (24)) with** $\Lambda_1 = 0.28$ **(day$^{-1}$) and** $E_{1,\infty_e} = 6.1$ **(units) (** $\lambda_{1,e}(E)$ **, red, dashed) and extended power law** $\lambda_w(E)$ **(Eq. (26)) with the exponent** $\sigma = 0.5$**, the coefficient** $a = 0.43$ **(units$^{0.5}$/day), and** $E_{1,\infty_w} = 6.6$ **(units) (** $\lambda_{1,w}(E)$ **, blue, dashed). (d) Error growth rate** $\lambda_\tau(E)$ **where** $\tau = tot$ **(black, thin),** $\tau = 1$ **(red, thin),** $\tau = 2$ **(orange, thin),** $\tau = 3$ **(blue, thin). Extended power law** $\lambda_{\tau,w}(E)$ **where** $\tau = tot$ **(black),** $\tau = 1$ **(red),** $\tau = 2$ **(orange) with** $E_{2,\infty_w}$ **(orange, dashdotted) ,** $\tau = 3$ **(blue) with** $E_{3,\infty_w}$ **(blue, dashdotted). Power law** $\lambda_{tot,p}(E)$ **(black, dashed). Extended exponential growth** $\lambda_{\tau,e}(E)$ **where** $\tau = 1$ **(red, dashed) with** $\Lambda_1$ **(red, dotted),** $\tau = 2$ **(orange, dashed) with** $\Lambda_2$ **(orange, dotted),** $\tau = 3$ **(blue, dashed) and** $\Lambda_{teor}$ **(black, dotted).**



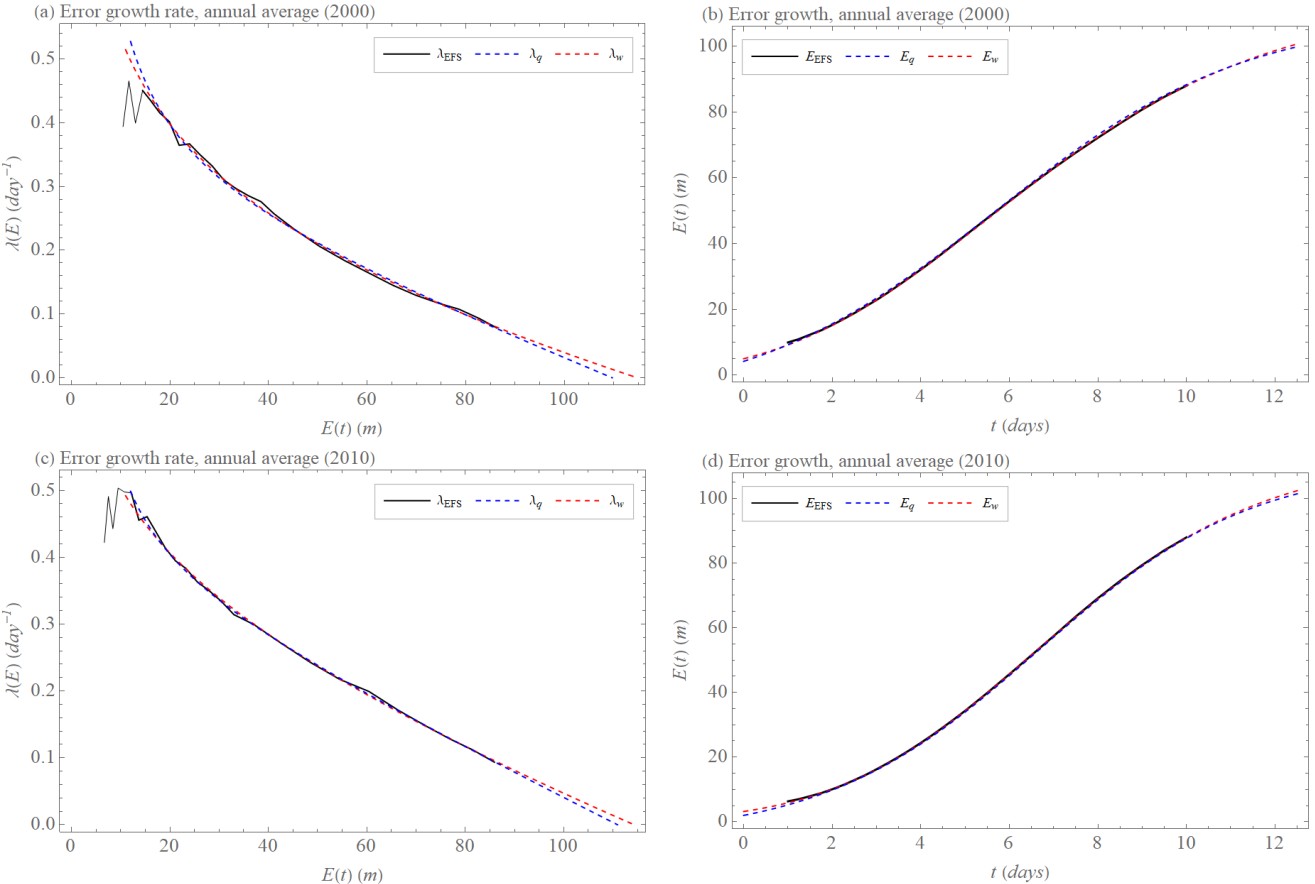

**Figure 5. The error growth rate** $\lambda_{EFS}(E)$ **and error growth** $E_{EFS}(t)$ **of the ECMWF forecasting system's 500 hPa geopotential height values (Northern Hemisphere (20°–90°)) calculated as differences between two operational forecasts issued with one day lag but valid on the same day (black), approximation by the extended power law** $\lambda_w(E)$ **(Eq. (26)) and** $E_w(t)$ **(Eq. (27)) (red, dashed), and approximation by the extended quadratic model** $\lambda_q(E)$ **(Eq. (20)) and** $E_q(t)$ **(Eq. (21)) (blue, dashed). (a,b) Annual average of 2000,** $\lambda_{EFS,w}(E)$ **and** $E_{EFS,w}(t)$ **with** $\sigma = 0.28$ **,** $a = 1.1$ **(m$^{0.28}$/day), and** $E_\infty = 115$ **(m),** $\lambda_{EFS,p}(E)$ **and** $E_{EFS,p}(t)$ **with** $\alpha = 0.32$ **(day$^{-1}$) and** $\beta = 3.2$ **(m/day) and** $E_\infty = 110$ **(m). (c,d) Annual average of 2010,** $\lambda_{EFS,w}(E)$ **and** $E_{EFS,w}(t)$ **with** $\sigma = 0.17$ **,** $a = 0.8$ **(m$^{0.17}$/day), and** $E_\infty = 114$ **(m),** $\lambda_{EFS,p}(E)$ **and** $E_{EFS,p}(t)$ **with** $\alpha = 0.4$ **(day$^{-1}$) and** $\beta = 2$ **(m/day) and** $E_\infty = 111$ **(m).**





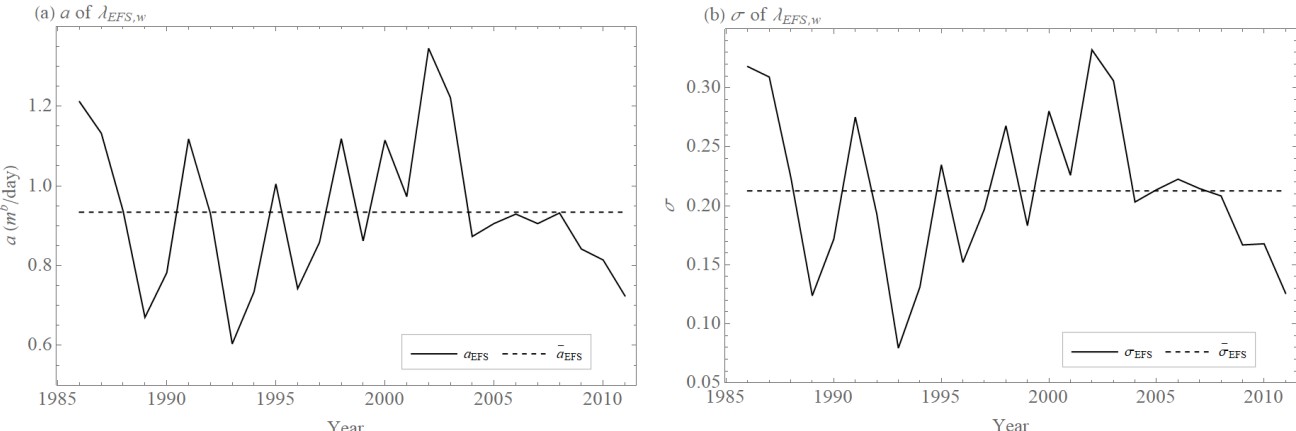

**Figure 6. Values of coefficients $a$ (a) and exponents $\sigma$ (b) of the extended power law $\lambda_w(E)$ (Eq. (26)) approximated from annual averages of the error growth rate $\lambda_{EFS}(E)$ of the ECMWF forecasting system's 500 hPa geopotential height values (Northern Hemisphere (20°–90°)) calculated as differences between two operational forecasts issued with one day lag but valid on the same day (black) and the average value of coefficients $\bar{a}_{EFS} = 0.93 \pm 0.18$ m$^\sigma$/day and exponents $\bar{\sigma}_{EFS} = 0.21 \pm 0.07$ over all years (black, dashed).**

515





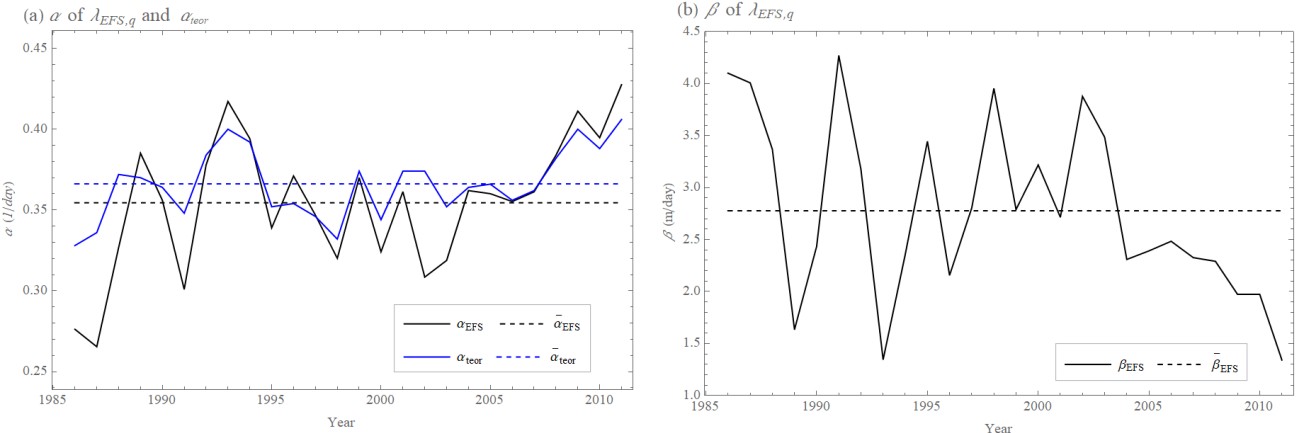

**Figure 7. Values of parameters** $\alpha$ **(a) and** $\beta$ **(b) of the extended quadratic model** $\lambda_q(E)$ **approximated from annual averages of the error growth rate** $\lambda_{EFS}(E)$ **of the ECMWF forecasting system's 500 hPa geopotential height values (Northern Hemisphere (20°– 90°)) calculated as differences between two operational forecasts issued with one day lag but valid on the same day (black), the average value of parameters** $\bar{\alpha}_{EFS} = 0.35 \pm 0.04$ **day$^{-1}$ and** $\bar{\beta}_{EFS} = 2.8 \pm 0.9$ **m/day over all years (black, dashed) and theoretical value of** $\alpha_{teor}$ **(blue) and its average value** $\bar{\alpha}_{teor} = 0.37 \pm 0.02$ **day$^{-1}$ (blue, dashed) over all years determined by Bednar et al. (2020).**

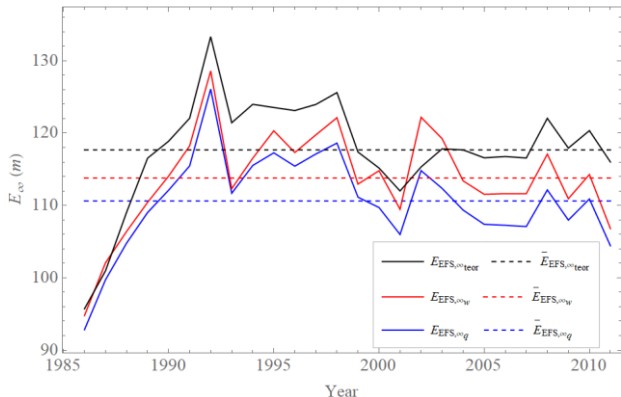

**Figure 8. Values of limit (saturated) values** $E_{EFS,\infty_w}$ **of the extended power law** $\lambda_w(E)$ **(red),** $E_{EFS,\infty_q}$ **of the extended quadratic model** $\lambda_q(E)$ **(blue), and theoretical value** $E_{EFS,\infty_{teor}}$ **determined by Bednar et al. (2020) (black) approximated from annual averages of the error growth rate** $\lambda_{EFS}(E)$ **of the ECMWF forecasting system's 500 hPa geopotential height values (Northern Hemisphere (20°–90°)) calculated as differences between two operational forecasts issued with one day lag but valid on the same day. The average value of** $\overline{E}_{EFS,\infty_w} = 114 \pm 7$ **m (red, dashed),** $\overline{E}_{EFS,\infty_q} = 111 \pm 7$ **m (blue, dashed), and** $\overline{E}_{EFS,\infty_{teor}} = 118 \pm 7$ **m (black, dashed).**



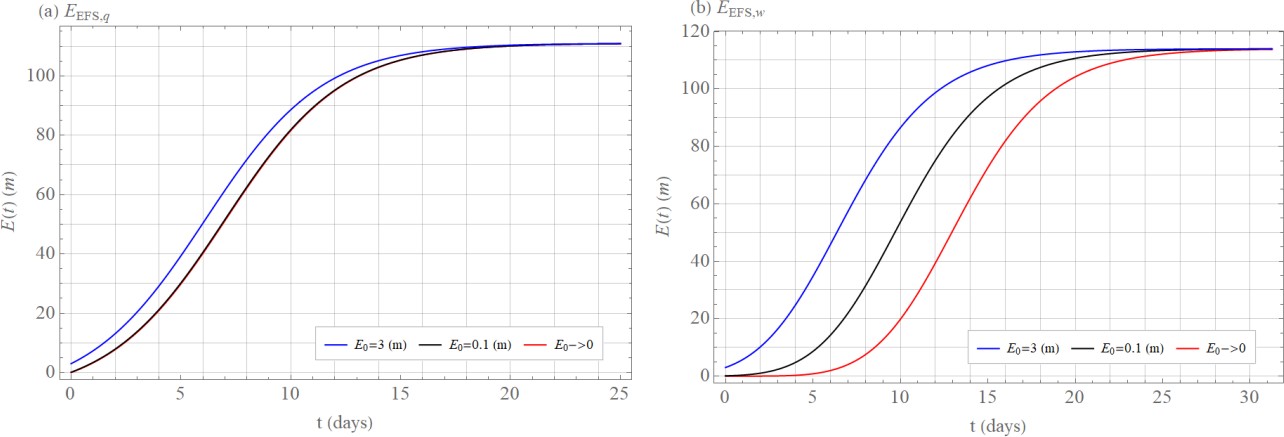

**Figure 9. (a) The solution of the extended quadratic model** $E_q(t)$ **with parameters** $\bar{\alpha}_{EFS} = 0.35$ **day$^{-1}$ and** $\bar{\beta}_{EFS} = 2.8$ **m/day, limit (saturated) value** $\bar{E}_{EFS,\infty_q} = 111$ **m and with initial errors** $E_0 = 3$ **m (blue),** $E_0 = 0.1$ **m (black), and** $E_0 \rightarrow 0$ **(red). (b) The solution of the extended power law** $E_w(t)$ **with the coefficient** $\bar{a}_{EFS} = 0.93$ **m$^\sigma$/day, exponent** $\bar{\sigma}_{EFS} = 0.21$**, limit (saturated) value** $\bar{E}_{EFS,\infty_q} = 114$ **m and with initial errors** $E_0 = 3$ **m (blue),** $E_0 = 0.1$ **m (black), and** $E_0 \rightarrow 0$ **(red).** ‾ **means averages over approximations of** $\lambda_q(E)$ **and** $\lambda_w(E)$ **from annual averages of the error growth rate** $\lambda_{EFS}(E)$ **of the ECMWF forecasting system's 500 hPa geopotential height values (Northern Hemisphere (20°–90°)) calculated as differences between two operational forecasts issued with one day lag but valid on the same day.**