# Peer review of "Prediction Error Growth in a more Realistic Atmospheric Toy Model with Three Spatiotemporal Scales"

_Geoscientific Model Development, 2021_

## Author Response (AR1)

**In response to RC1**, we clarified that an infinite Lyapunov exponent would be an idealized limit case of a power law error growth, which for real physical systems has a lower cut-off leading to a very large but finite Lyapunov exponent.

In response to RC1, we inserted a short subsection 3.1 on scale dependent error growth and how to measure it numerically, where we refer to the literature in more detail. There we discuss why we do not use the strictly scale dependent finite size Lyapunov exponents but instead couple error magnitude and evolution time, as outlined in the response letter to RC1. We also added some lines to the introduction containing references to literature about prediction errors and Lyapunov exponents.

We repeated the error growth analysis for our model using the proper definition of the finite size Lyapunov exponent. The results are qualitatively in full agreement with those of our type of analysis for the model system, with small quantitative differences. Since one cannot compute this FTLE for the ECMWF data, and since our approach is in close analogy to the forecasting issue, we did not include the FTLE results in the revised version of the paper, but we added a corresponding comment to the end of Section 3.2.

We corrected the formula in line 257.

We add a comment of why our numerical analysis of error growth rates does not show the convergence to the true Lyapunov exponent in the conclusions, reacting to RC1.

**Response to referee 1**
We are grateful to the referee for devoting their time to our manuscript. The valuable comments and suggestions will help us to improve the paper.

We will here respond to the main comments made:

1) We are very sorry for not citing all of the relevant literature on scale dependent error growth, in particular in turbulence, and how to measure it numerically. We will fix this in the revised version.
Concerning the calculation of the error growth rates, we were indeed using a different scheme than proposed in Aurell et al PRL 77 1262 (1996): We create a perturbed field close to the reference field, we iterate them both for some transient time to allow the perturbation to re-direct into the locally most unstable direction, and then track the increase of the perturbation in time. Then (Fig.2a) we average over the error growth rate at given times after initialization, and plot it versus the mean error magnitude at that time (called $E(t)$ in Fig.2a). The reason for doing so is that in real forecasts, it is standard to study the average error after some given time. In particular, when performing the analysis for the ECMWF ensemble forecasts, we are only evaluating error growth experiments which have been performed by others, so that we do only have access to the error growth rate after fixed times, and not at given error magnitudes.
We fully agree that the way how the averages are obtained generally will have an influence on the numerical results. We have therefore now repeated the error growth analysis for our model system in the way of Aurell et al. Qualitatively, the results are the same. With our method, the power law behavior $\lambda(E) \propto E^{-\sigma}$ is slightly better than for Aurell's method, which seems to have the tendency to resolve the Lyapunov exponents of the different levels. This smoothing of the error growth rate most certainly comes from the fact that, as suspected by the referee, our method mixes the scales: In the average over many error growth experiments, at some given time $t$ after initialization, the actual errors $E(t)$ have different magnitudes. In addition, we observe a slightly lower error growth rate for the same value of error magnitude, which we understand also by this mixing of scales: If $\lambda(E) \propto E^{-\sigma}$, then $\langle \lambda(E) \rangle < \langle E \rangle^{-\sigma}$, where the average is assumed to be done over the error magnitudes which are found after fixed time $t$.

So our conclusion is that although a scale dependent error growth per se should be studied in the way of Boffetta et al, a comparison with real weather forecast data can only be done by studying the time evolution of errors. We will add a corresponding discussion in the revised version of the manuscript.

2) line 33-34: Our statements about the "true" largest Lyapunov exponent being infinite were imprecise. They refer to a model where the scale dependent Lyapunov exponent reads $\lambda(\varepsilon) \propto \varepsilon^{-|\sigma|}$ and hence $\lambda \to \infty$ for $\varepsilon \to 0$. Evidently, such a behavior would be an idealization (approximation) since in real systems like in turbulence one would expect to have some cut-off at some small length- scales with a cross-over to some finite $\lambda_{max}$. A comment like this was made in the publication Brisch & Kantz to which we refer in the manuscript, but I agree that we have to include it here as well.

3) We are sorry for ignoring relevant literature on scale dependent error growth and will give adequate merits to that in the revised version.

4) line 225 (257): We fully agree, we erroneously forgot a prefactor $1/\ln \varepsilon$. We are grateful for pointing out this mistake.

5) Yes, we are talking here about the scale dependent error growth on scales which are already in the nonlinear regime. Due to the initial transient after the initialization of the perturbation, we do not observe the classical exponential error growth on very small scales, but it should be there. We will explore whether by the choice of even smaller perturbations, we can reach this regime numerically.

6) Unfortunately, it is true that for the ECMWF model results, we can not clearly distinguish between our conjecture of scale dependent error growth and a previous suggestion of how to fit it and how to interpret it. We have to understand better whether this is a consequence of (in term of scale dependent error growth) too coarse scales of the EMCWF model or of too large initial errors in the error growth experiments of ECMWF.

Concerning the referee's conclusion we agree that our study of scale dependent error growth is not exactly in the spirit of FSLE. Inspired by the referee's comments, we calculated the FSLE for our model, and we are considering to include the results in the revised version. However, the results of our "time after perturbation" approach do not differ qualitatively, and this is the only approach which works for the ECMWF model without modifying ECMWF's code and running this model ourselves, which is out of reach to us.

**In response to RC2**, we included in the abstract that a fit of the quadratic law to the ECMWF data gives a shorter prediction horizon of 15 days, while a fit of the extended power law yields 22 days. Other modifications are described in the text.

**Response to referee 2**
We are grateful to the referee for devoting their time to our manuscript. The valuable comments and suggestions will help us to improve the paper.

We will here respond to the main comments made:

*In the abstract, the authors wrote there is an intrinsic limit of predictability after 22 days, this conclusion is made by fitting the modified power law function to the ECWMF data. If the function form in Zhang et al 2019 is used, then the predictability limit becomes 15 days as mentioned in the text. The difference between these two estimated limits is sort of large. In the context of the ECMWF operational forecast system, the reviewer did not find any advantage of using the modified power law function rather than the function form in Zhang et al. Is there any reason for the reader to believe that this 22-day limit is more accurate than the 15-day limit?*

This is indeed a valid and relevant question. As we write in our manuscript (lines 449-458), from the data of error growth experiments performed with the ECMWF forecast system, it cannot be decided which law for the error growth is more appropriate. We provided some evidence that the extended power law has some theoretical justification and matches well the observations from our toy-model. The two competing error growth laws fitted to the ECMWF data yield different results for the prediction horizon,

namely 15 versus 22 days. Due to the theoretical justifications behind the extended power law, we are inclined to follow its results and to be optimistic. Please also consider that the current forecasts do lose their skill after about 10-15 days, and that the quadratic law is a fit to such forecasts without extrapolation to smaller initial condition errors. Our 22 days are an upper bound for the prediction horizon for hypothetical forecasts with perfect initial conditions. Also, these 22 days are in nice agreement with Krishnamurthy (2019).

*Scale-dependence is the key for understanding atmospheric predictability. The authors proposed this three-scale toy model. Could it be possible to verify this three-scale model with the ECMWF data? E.g., to connect $X_1$ with synoptical errors, $X_2$ with meso-scale error and $X_3$ with turbulent motions. If such filters are applied to the ECMWF data and verify the errors of different scales with the toy model, then the results would be much more convincing.*

We agree that it would be nice to build such a three-scale model from the ECMWF model. However, we do not think that reality has only 3 scales. The atmosphere has a large variety of spatial and temporal scales, and even a cross- over from essentially 2-dimensional transport on very large spatial scales to 3-d on small scales. Also, from the practical point of view, we do not know how to do that: We could adapt and then apply our filters to the ECMWF model in order to construct 3 data sets representing different scales. But that would require a suitable fine-tuning of parameters in our filters, and therefore would be a project of its own which will be worth to be started. Our toy model only represents something like a latitude circle and not the globe. Hence, our toy-model is much simpler than any atmospheric model and serves as a kind of didactical example of for multi-scale models. However, we have a set of similarities with real atmospheric models which we want to stress here again:
1. Our variable $X_1$ has 5 to 7 main highs and lows that correspond to planetary waves (Rossby waves) and several smaller waves corresponding to synoptic-scale waves.
2. Parameters of our three-scale toy model are chosen in order for the medium scale ($X_2$) amplitude to be approximately ten times smaller than the large scale amplitude and the small scale amplitude ($X_3$) again approximately ten times smaller than the medium scale amplitude. Our variables also have different oscillation periods (lines 176 -179), which can be roughly compared to synoptic, meso, and turbulent scales (lines 48 - 56).
3. Bednar (2020) showed a similarity in the error growth of one-scale toy model (Eq. (2)) and ECMWF data.

*Once the parameter of the three-scale toy model is set, then its error growth behavior is also determined, Is the results shown here sensitive to the value of the parameters in the equation (e.g. F, b, c, I) ?*

We have chosen parameters such that all levels behave chaotically (the largest Lyapunov exponent of each level is positive) and that all levels have a significant difference in amplitudes and fluctuation rates (lines 160 - 162). For such conditions, the general results are not sensitive to parameter values in quality, but the values of fit-coefficients of the different error growth laws will differ.

*Line 14-15: a theoretical justification of function form in Zhang et al. is recently provided by Sun and Zhang 2020 (https://doi.org/10.1175/JAS-D-19-0271.1).*

We are grateful for pointing out this reference, of which we were not aware. We changed lines 14 - 15 to: Although the quadratic hypothesis cannot be completely rejected and could serve as a first guess, the hypothesis parameters are not theoretically justifiable in the model. We added the mentioned citation: Lines 379; 381; 439 and 517 – 518.

*Line 180 (207): perfect model assumption is used, right?*
Yes, we added this to our text (Line 207).

*Line 214: what does this initial transient behavior look like? Is the error de- creasing with time? What would happen if the initial error is further reduced towards 0?*

Yes, before the perturbed trajectory has relaxed back to the attractor, the error is decreasing with time. When the magnitude of the initial perturbation is reduced, the relaxation time of the perturbed trajectory also decreases. We were using initial error magnitudes which were optimized for having minimal but non-zero errors after the end of the transient.

*A bracket is missing in Equation. (20)*
Many thanks, we corrected this.

*Line 365 (396): the extended quadratic model Eq. (21) is more accurate than the extended exponential growth? How to reconcile this with the three-scale toy model results?*

Originally, the extended exponential error growth was designed to describe the error growth and error growth rate in 1-dimensional models, where "extended" means that it also captures the saturation of error growth on large scales. So the extended exponential model was made for a scale-independent error growth rate, while for our 3-scale model, we had implemented the "extension" to saturation into the power law error growth. Since the ECMWF systems exhibits multi-scale dynamics with scale dependent error growth, the extended exponential error growth model is less accurate than the extended quadratic model, but it was our intention to compare the observed error growth to the extended power law. Bednar et al. (2020) showed that the extended quadratic model Eq. (21) is more accurate than the extended exponential growth for ECMWF data, but in that work the extended power law was not tested.

*Line 415-425 (449 – 458): The authors seem to hint that the extended power-law form is better compared to the extended quadratic form here. This is true for the toy model. But it is not supported by the ECMWF, right?*

Yes, this is right, for the ECMWF model, both fits are of similar quality. In lines 449 - 458 we argue what we can conclude if we assume that nonetheless the extended power law is the correct description of the data.

---

## Author Response (AR2)

**Response to referee:**

We are grateful to the referee for devoting time to our manuscript and we thank the referee for suggestions. We fixed small grammar details and improved the text exposure. Specially:

**Lines 32-34:** We checked the text, fixed some details, and improved clarity.

**Lines 188-202:** We thank the referee for drawing our attention to the FSLE and fully agree with the referee's view.

**Lines 247-252:** We deleted the last sentence of the paragraph.

---

## Author Response (AR3)

**Response to referee:**

We are grateful to the referee for devoting time to our manuscript, and we thank the referee for suggestions. We fixed typos and other errors. The changes are contained in the track changes part (further in the document). Specially:

*L125: Is there an F term missing in eq 6? or it is not identical to eq 1:*

[revised manuscript text omitted]